# Evaluation of the Sensitivity of Breast Cancer Cell Lines to Cardiac Glycosides Unveils ATP1B3 as a Possible Biomarker for the Personalized Treatment of ERα Expressing Breast Cancers

**DOI:** 10.3390/ijms231911102

**Published:** 2022-09-21

**Authors:** Filippo Acconcia

**Affiliations:** Department of Sciences, University Roma Tre, Viale Guglielmo Marconi, 446, I-00146 Rome, Italy; filippo.acconcia@uniroma3.it; Tel.: +39-0657336320; Fax: +39-0657336321

**Keywords:** breast cancer, estrogen receptor α, ouabain, cardiac glycosides, Na/K ATPase, ATP1B3, personalized medicine

## Abstract

The molecular classification of breast cancer (BC) dictates pharmacological treatment. Estrogen receptor α (ERα) expressing tumors are treated with 4OH-tamoxifen or fulvestrant, which inhibits the receptor, or with aromatase inhibitors (i.e., anastrozole, letrozole, and exemestane) that reduce the 17β-estradiol (E2) circulating blood levels. Besides such endocrine therapy (ET) drugs, ERα-positive BCs can be treated with epidermal growth factor receptor (EGF-R) inhibitors (i.e., gefitinib, erlotinib, and lapatinib) according to HER2 expression. Notwithstanding these anti-BC drugs, novel personalized approaches for BC treatment are required because prolonged administration of those pharmaceutics determines resistant phenotypes, which result in metastatic BC. We have recently reported that the cardiac glycoside (CG) (i.e., Na/K ATPase inhibitor) ouabain could be repurposed for ERα-positive primary and metastatic BC treatment as it induces ERα degradation and kills BC cells. Here, we evaluated if other CGs could represent additional treatment options for ERα-positive BCs and if the Na/K ATPase could be considered a biomarker for ERα-positive BC treatment. The results indicate that the ATP1B3 Na/K ATPase isoform can educate the choice for the personalized treatment of ERα-positive BC with CGs and that CGs could be more efficacious if they are administered in association with gefitinib.

## 1. Introduction

Different molecular subtypes of breast cancer (BC) exist and are called clinical surrogates. The molecular classification of BC is necessary to drive the therapeutic approach for this disease. Indeed, most of BCs express the estrogen receptor α (ERα) and display a more favorable prognosis with respect to ERα-negative tumors that are often more aggressive and determine an unfavorable outcome [1].

The ERα expression dictates the clinical treatment of BCs [1,2]. Women carrying ERα-positive tumors are, in general, treated with endocrine therapy (ET) drugs [2]. 4OH-tamoxifen (Tam), which is the prototype selective estrogen receptor modulator (SERM), and fulvestrant (Ful, also known as ICI182,780–ICI), which is the prototype selective estrogen receptor down regulator (SERD), act by binding to the ERα and inhibiting receptor activities, which sustain BC cell proliferation [1,2]. The aromatase inhibitors (AI) such as anastrozole, letrozole, and exemestane are another class of ET that dampen the activity of the aromatase and consequently the production of the sex steroid hormone 17β-estradiol (E2) [2]. AI-dependent reduction of E2 levels in the bloodstream of ERα-positive patients diminishes BC progression because it removes the proliferative signaling induced by this sex steroid hormone [2]. The molecular classification of BC has revealed that ERα-positive tumors co-expressing the HER2 receptor (i.e., the luminal B (LumB) BCs) are less sensitive to the ET drugs than the ERα-positive tumors devoid of HER2 (i.e., the luminal A (LumA) BCs). In turn, women with LumB tumors are additionally treated with specific HER2 inhibitors (i.e., gefitinib, lapatinib, and erlotinib) [1].

Despite the proven efficacy of the treatments for ERα-positive BCs, the prolonged administration of women (i.e., 5 to 10 years) with these drugs often results in the development of resistant phenotypes and determines a relapse of the disease [1,2]. Unfortunately, such metastatic BC (MBC) patients become insensitive to drug treatment and, in most cases, carry a fatal disease [3]. Therefore, it is imperative to identify molecules that could complement the repertoire of anti-ERα-positive BCs drugs and that could work either alone or in combination with the already used pharmaceutics with the combined goal of both reducing the development of resistant tumors and treating them.

In this respect, we have shown that drugs inducing ERα degradation in BC cells through the activation of various intracellular mechanisms could block BC cell proliferation [4,5,6,7,8,9,10,11,12]. In this way, such drugs could provide additional opportunities for BC treatment because they can work alone or in a synergic manner with the drugs already used for the treatment of BCs [4,5,6,7,8,9,10,11,12]. Accordingly, we have recently reported that the cardiac glycosides (CGs) digoxin (Digo) and ouabain (OU) (i.e., two inhibitors of the Na/K ATPase) reduce ERα levels and block the proliferation of cell lines modeling primary and metastatic LumA BC cells [4]. Moreover, the analysis of their gene signature in LumA BC cells revealed that they reduce those genes that are overexpressed in Tam-resistant patients, thus suggesting that CGs could be effectively used for the treatment of LumA primary and metastatic tumors [4].

Because CGs are a class of numerous natural compounds with anti-tumor activity [13], here we sought to understand both if other CGs could represent additional treatment options for ERα-positive BC treatment to be administered alone or in combination with other anti-BC drugs, and if the Na/K ATPase could be considered a biomarker for addressing CG treatment in women carrying ERα-positive BC.

## 2. Results

### 2.1. Evaluation of the Sensitivity of Cardiac Glycosides in ERα-Positive Breast Cancer Cell Lines

We initially evaluated the sensitivity in terms of cell proliferation of BC cell lines to CG by using the DepMap portal (https://depmap.org/portal (accessed on 1 July 2022)). This free web-based database contains experimental data regarding the profiling of different parameters including drug sensitivity and other several omics data, such as the expression and protein array data in several cancer cell lines. The sensitivity of all the BC cell lines to all the CGs profiled in the database (i.e., resibufogenin (Resi), K-strophanthin (K-Stro), digitoxin (Digi), and digoxigenin (Digoxig), lanatoside-C (Lana), peruvoside (Peru), digoxin (Digo) and ouabain (OU)) was studied by stratifying the results according to the ERα expression [14,15]. Overall, ERα-positive BC cell lines are significantly more sensitive to CGs than the ERα-negative BC cell lines (Figure 1A and Appendix A). While no differences have been observed between the sensitivity of ERα-positive and ERα-negative BC cell lines to Resi, K-Stro, Digi, and Digoxig, significantly higher sensitivities in ERα-positive BC cell lines than in ERα-negative BC cell lines (Figure 1B and Appendix A) have been scored for Lana, Peru, Digo, and OU.

We have previously demonstrated that the OU and Digo anti-proliferative activity is dependent at least in part on their ability to induce ERα degradation [4]. Therefore, we next interrogated the DepMap portal to understand if the higher sensitivity of the CGs Lana, Peru, Digo, and OU detected in ERα-positive BC cell lines (Figure 1B and Appendix A) could be correlated with ERα expression. As shown in Figure 1C–F, only OU showed a positive linear correlation (r = 0.6175; *p* = 0.043) with ERα mRNA expression (Figure 1F and Appendix A). Next, we further inspected if this trend could be different in BC cell lines considered as a model system for LumA (i.e., ERα-positive; PR-positive/negative and HER2-negative) or LumB (i.e., ERα-positive; PR-positive/negative and HER2-positive) BC [14,15]. As shown in Figure 1G,H (see also Appendix A), while no correlation between the sensitivity to OU and ERα mRNA expression was found in LumA BC cell lines, in LumB BC cell lines, the sensitivity to OU linearly increased with ERα mRNA expression (r = 0.9408; *p* = 0.0592).

Altogether, the reported observations indicate that ERα-positive BC cell lines are more sensitive to CGs than ERα-negative BC cell lines and that the sensitivity of ERα-positive BC cell lines to OU increases in cells expressing high ERα mRNA levels.

### 2.2. Evaluation of the Effects of Ouabain in Cells Modeling Luminal B Breast Cancer

Because we observed that in cells modeling LumB BC the sensitivity to OU increases with the ERα mRNA expression, the effects of OU in the BT-474 cell line, which represent a cellular system for triple-positive BC (i.e., LumB) [14,15], were next evaluated.

Growth curve analyses showed that OU and fulvestrant (ICI182,780—ICI), used as an internal control, reduce the BT-474 cell proliferation. Interestingly, the effect of OU in preventing BT-474 cell proliferation is higher than that elicited by ICI administration (Figure 2A). Next, we evaluated the impact of OU on the cell cycle. According to previous results [4], 24 h OU administration to BT-474 cells significantly increased the percentage of the cells in the G2 phase of the cell cycle (Figure 2B). A colony formation assay was further performed to evaluate the ability of BT-474 cells to survive and form a colony in the presence of OU. Additionally, in this case, ICI was used as the internal control. As shown in Figure 2C, treatment of BT-474 cells for 3 weeks with either OU or ICI significantly prevented the ability of the cells to form colonies.

Because we previously demonstrated that OU induces ERα degradation in cell lines modeling LumA primary (i.e., MCF-7 and ZR-75-1) and metastatic BC (i.e., MCF-7 Tam Res and expressing the Y537S receptor mutant) [4], the effect of OU on the regulation of the receptor in BT-474 cells was next evaluated. As shown in Figure 2D,D’, 24 h OU administration induced a dose-dependent reduction in ERα intracellular levels. Finally, we measured the ability of OU to inhibit the Na/K ATPase activity in BT-474 cells treated for 24 h with this CG. As shown in Figure 2E, as expected, OU inhibited the Na/K ATPase with an inhibitory concentration 50 (IC_50_) of 1150 ± 430 nM. Remarkably, in BT-474 cells, the calculated IC_50_ of OU for the inhibition of cell proliferation (Figure 2F) and the effective concentration 50 (EC_50_) for its ability to reduce receptor levels (Figure 2D) were 3.3 ± 0.4 nM and 3.3 ± 1.7 nM, respectively. This evidence suggests that the inhibition of Na/K ATPase by OU is independent of the ability of this CG to induce receptor degradation and prevent cell proliferation. As we previously showed that OU activates the 26S proteasome activity in LumA primary and metastatic BC cells, but not in normal breast epithelial cells [4], we tested if this mechanism could be active also in BT-474 cells. As shown in Figure 2F,F’, the treatment of BT-474 cells with OU reduced in a dose-dependent manner the total amount of the ubiquitinated species. Moreover, 24 h administration of OU to BT-474 cells increased the 26S proteasome activities (i.e., chymotrypsin-like, caspase-like, and trypsin-like activity) in a dose-dependent manner (Figure 2G). As expected, the 26S proteasome inhibitor Mg-132 effectively inhibited the 3 proteasome activities (Figure 2G’). Next, we treated cells with Mg-132 both in the presence and in the absence of OU. Mg-132 administration to cells increased the amount of total ubiquitinated species, and OU was able to reduce them both in the presence and the absence of Mg-132 (Figure 2H,H’). As a control, OU dose–response analyses were performed on cell growth (Figure 2I), Na/K ATPase activity (Figure 2L), 26S proteasome activity (Figure 2M,M’), and total content of cellular ubiquitinated species (Figure 2O,O’) in MDAMB231 cells, which are triple negative cell lines of basal origin [14,15]. Data show that OU-dependent activation of the proteasome activity is inconsistent in MDAMB231 cells and accordingly OU is less efficacious than in BT-474 cells in reducing total ubiquitinated species. Moreover, the calculated IC_50_ of OU for the inhibition of cell proliferation and Na/K ATPase activity were 29.8 ± 3.7 nM and 13.8 ± 1.5 nM, respectively. These data suggest that in triple negative BC cells MDAMB231, the OU effect could occur because of the inhibition of the Na/K ATPase activity rather than the inhibition of 26S proteasome activity.

Overall, the data reported here indicate that OU activates the 26S proteasome, prevents cell proliferation, and induces ERα degradation in cells modeling the LumB BC phenotype, thus demonstrating that OU works in LumB ERα-positive BC cell lines as we already reported in two LumA BC cell lines (i.e., MCF-7 and ZR-75-1 cells) [4] and further suggest that the increased anti-proliferative activity of this CG in ERα-positive cell lines with respect to ERα-negative cell lines could be due to OU-induced receptor degradation and 26S proteasome activation.

### 2.3. The Impact of Na/K ATPase in Breast Cancer Progression

Prompted by these results, we next reasoned that the expression of Na/K ATPase, which is the pharmacological target of OU, could dictate a different survival rate in women carrying ERα-positive or ERα-negative breast tumors.

To tackle this issue and because different Na/K ATPase isoforms are expressed in different human tissues [16], we initially evaluated all the isoforms (i.e., ATP1A1, ATP1A2, ATP1A3, ATP1A4, ATP1B1, ATP1B2, ATP1B3, and ATP1B4) the mRNA expression levels in breast tumors and normal breast tissue by using the public available TNMplot database (https://tnmplot.com/analysis/ (accessed on 1 July 2022)) and used the RNAseq data [17]. As shown in Figure 3A (see also Appendix A), ATP1A1, ATP1A2, ATP1B1, and ATP1B3 Na/K ATPase isoforms displayed a high mRNA expression in normal breast tissue with respect to the other isoforms. Interestingly, ATP1A1 and ATP1B1 isoforms were overexpressed in breast tumors while ATP1A2 was downregulated in breast tumors and ATP1B3 mRNA expression was unchanged (Figure 3A).

Next, we sought to understand whether different levels of ATPA1, ATPA2, ATPB1, and ATPB3 mRNA expression could impact the survival of women carrying ERα-negative or ERα-positive BCs to gain insight into the possibility to use OU or possibly other CGs as an additional treatment option for tumor management. Kaplan–Meier curves were retrieved by the Kaplan–Meier Plotter database (https://kmplot.com/analysis/ (accessed on 1 July 2022)) [18]. Women with ERα-negative BC display a significantly increased relapse-free survival (RFS) only when the tumor expresses low levels of ATP1A1 (Figure 4A,C,E,F; see also Appendix A). On the contrary, women with ERα-positive BC have a significantly higher survival probability when the tumor expresses high levels of ATP1A1 or ATP1B1 (Figure 4A,F; see also Appendix A) while no significant changes in RFS of women with ERα-positive BC have been evidenced considering the ATP1A2 mRNA expression (Figure 4D; see also Appendix A). Surprisingly, women expressing low levels of ATP1B3 mRNA have a higher survival probability with respect to those expressing high levels of ATP1B3 (Figure 4H; see also Appendix A). Overall, these data indicate that an increased expression of ATP1A1 and ATP1B1 in women carrying ERα-positive tumors does not indicate the possibility to use OU or other CGs and suggests that OU or other CGs administration could be an exploitable option in ERα-positive BC in those women expressing high levels of ATP1B3.

To test this hypothesis, we inspected the DepMap portal to evaluate the potential correlation between the sensitivity to OU and the mRNA expression of ATP1A1, ATP1A2, ATP1B1, and ATP1B3 in ERα-positive BC cell lines. According to the RFS data, we observed a significant positive linear correlation between the sensitivity to OU and ATP1B3 (Figure 5D; see also Appendix A), but not with ATP1A1, ATP1A2, and ATP1B1 (Figure 5A–C; see also Appendix A).

Because ATP1A1 and ATP1B1 mRNA expression is upregulated while ATP1A2 expression is down-regulated, we next determined their differential expression in ERα-positive BC and ERα-negative BC by using the Metabric data available in the https://www.cbioportal.org/. (accessed on 1 July 2022) database. As a control, we also included the analysis of ATP1B3 as its mRNA expression is unchanged between normal and tumor tissue.

As shown in Figure 3B (see also Appendix A), no differences have been detected in the mRNA expression levels between ERα-positive and ERα-negative breast tumors for ATP1A1 and ATP1A2 while an increased mRNA expression has been detected for ATP1B1 in ERα-positive than in ERα-negative breast tumors. On the contrary, a reduced mRNA expression was scored in ERα-positive than in ERα-negative breast tumors for ATP1B3 (Figure 3B).

### 2.4. Evaluation of Potential Synergy between Ouabain and the Drugs Used to Treat ERα-Positive Breast Cancers

We have previously [4] demonstrated that OU shows synergic anti-proliferative effects with 4OH-tamoxifen (Tam) in two cellular systems modeling LumA BC (i.e., MCF-7 and ZR-75-1 cells) [14,15]. Present results demonstrate that OU also has anti-proliferative activity in cells modeling LumB BC (i.e., BT-474 cells) and that the ERα-positive BC cell lines expressing high levels of ATP1B3 are highly sensitive to OU anti-proliferative effects.

Therefore, we finally sought to determine whether OU treatment could synergize with the drugs used in the clinics to treat ERα-positive tumors (i.e., fulvestrant, AI (anastrozole, letrozole, and exemestane) and HER2 inhibitors (erlotinib, lapatinib, and gefitinib)). For this purpose, we retrieved the sensitivity data from the DepMap database and performed correlation analyses. Significant inverse linear correlation has been evidenced in ERα-positive BC cell lines for the sensitivity to OU and fulvestrant (Figure 6A; see also Appendix A) or OU and letrozole (Figure 6C; see also Appendix A) while the direct significant correlation was found in the sensitivity of ERα-positive BC cell lines between OU and gefitinib (Figure 6G; see also Appendix A). No significant correlations were detected between the sensitivity to OU and that of the other drugs (Figure 6B,D–F; see also Appendix A).

Overall, these data suggest that selective co-administration of OU with specific ET drugs in ERα-positive BC could provide possible additional strategies for the management of this kind of disease.

## 3. Discussion

Despite the progress in the therapy for breast cancer (BC), this disease remains the most frequent and deadly neoplasm for women [19]. At the diagnosis, the molecular classification of breast tumors is necessary to drive the most appropriate therapeutic strategy, which is based on the specific molecular features of the tumor [1,2]. Thus, BC patients already have the advantage of personalized treatment for the disease. Indeed, the presence of the ERα dictates the use of endocrine therapy (ET) and the sub-classification of ERα-positive tumors into luminal A (LumA) (i.e., devoid of HER2) or LumB (i.e., expressing HER2 together with ERα) further indicates the use of chemotherapeutic agents, as LumB tumors are less sensitive than LumA tumors in terms of the efficacy of ET [1,2]. However, prolonged treatment with anti-BC drugs often determines the development of resistant phenotypes for which the initial treatment is ineffective [20,21,22]. In turn, additional therapeutic strategies are required to refine the personalized treatment of ERα-positive tumors with the goal either to avoid the development of resistant phenotypes or to treat resistant tumors.

Our laboratory has introduced a novel perspective to identify new anti-BC drugs by implementing the degradation of the ERα as a primary target for treatment [11]. Indeed, although different molecular mechanisms exist for the development of resistant tumors [11,20,21,22], they remain addicted to the proliferative activities driven by the ERα, which as a transcription factor, regulates the expression of the genes required for cell proliferation in both the primary and the metastatic tumor context [23]. In this respect, in the recent past, among other drugs [4,5,6,7,8,9,10,11,12], we have identified that the CGs OU and Digo prevent the proliferation of LumA cell lines modeling primary and metastatic BC by hyper-activating the 26S proteasome, which in turn determines the degradation of the ERα, the induction of the apoptotic cascade and, consequently, cell death [4]. These discoveries have led us to propose CGs as potential novel anti-BC drugs that could work as “anti-estrogen-like” pharmaceutics [4].

In this work, through a combination of bioinformatics and molecular biology approaches, we report that ERα-positive BC cell lines are overall more sensitive to the antiproliferative effects of CGs than the ERα-negative cell lines, most likely because these compounds can activate the 26S proteasome and induce the ERα degradation, thus increasing their anti-proliferative activity in ERα-positive BC cell lines. According to this notion, we found that the sensitivity to OU of ERα-positive BC cell lines is linearly correlated with the receptor mRNA expression. Furthermore, such correlation is more evident in cell lines modeling LumB tumors. Interestingly, because LumB BC generally expresses a lower level of ERα than LumA BCs, the present observation supports the concept that the more the receptor is expressed the more OU has an anti-proliferative activity. In turn, we extend our previous findings regarding the effect of OU in ERα-positive BC cells from LumA cell lines [4] to LumB cell lines. Indeed, also in BT-474 cells, which model LumB tumors [14,15], OU administration induces both ERα degradation, 26S proteasome activation, and prevents cell proliferation. Therefore, the present results together with those previously reported [4] indicate that OU prevents the growth of cells modeling both LumA (i.e., MCF-7 and ZR-75-1 cells), LumB (i.e., BT-474 cells), and metastatic ET-resistant BC cells (i.e., MCF-7 Tam Res and expressing the Y537S receptor mutant). Additionally, the observations extracted from different BC cell lines together with the fact that OU induces receptor degradation implicate that the increased sensitivity to OU of ERα-positive BC cell lines with respect to ERα-negative BC cell lines could be due to the ability of OU to eliminate ERα. Therefore, OU, and possibly other CGs, could be re-purposed as treatments for ERα-positive breast tumors.

In the perspective of providing indications for the personalized uses of OU as a function of the molecular phenotypes of BC, we next evaluated the ERα-positive BC progression in relationship with the expression of the Na/K ATPase, which is the pharmacological target for OU and other CGs [13]. Eight Na/K ATPase isoforms exist (i.e., ATP1A1, ATP1A2, ATP1A3, ATP1A4, ATP1B1, ATP1B2, ATP1B3, and ATP1B4) and are differentially expressed in human tissues [16]. Among them, we found that ATP1A1, ATP1A2, ATP1B1, and ATP1B3 are highly expressed in normal breast tissue and among them, ATP1A1 and ATP1B1 are upregulated in breast tumors while ATP1A2 is downregulated and ATP1B3 is unchanged. In ERα-positive BC, instead, only ATP1B1 is upregulated, while ATP1B3 is downregulated with respect to ERα-negative BCs. Because, based on expression data, it is difficult to predict which isoform could be important for BC progression, we next analyzed the survival probability of women carrying ERα-negative or ERα-positive BC as a function of the mRNA expression of these four isoforms. Surprisingly, we found that an increased relapse-free survival probability for women is apparent when ERα-positive tumors express low mRNA levels of the ATP1B3 isoform, whereas women with ERα-positive tumors expressing low mRNA levels of ATP1A1 and ATP1B1 have a reduced survival probability.

In this respect, it has to be pointed out that past and present discoveries indicate that OU, in addition to the inhibition of the Na/K ATPase, can induce the hyperactivation of the proteasome, which acts by ERα inducing degradation [4]. These two effects occur in parallel in cancer cells treated with high doses of OU while at low doses of the cardiac glycoside the induction of receptor degradation prevails on the inhibition of the Na/K ATPase activity. In turn, the present data do not indicate that the OU pharmacological target is different from the Na/K ATPase rather they suggest that this drug has also the additional ability to hyperactivate the proteasome at least in the ERα-positive BC cells. In line with this evidence, we observed that in the triple-negative BC cells used in this study OU failed to activate the 26S proteasome. Therefore, overall, ERα-positive BC cells have a higher sensitivity to OU and possibly to other cardiac glycosides than ERα-negative BC cells. Interestingly, according to the pharmacological indications for human administration of OU, this glycoside is administered by slow intravenous infusion at a loading dose of 15µg/kg to be divided into three times at 2 h intervals as OU has a very low gastrointestinal absorption rate and rapid excretion rate [13,24,25,26]. Considering the OU molecular weight and a standard human being of 70 Kg containing 5 L of blood, these doses correspond to about 0.5 µM OU for each administration. At these doses, both the Na/K ATPase is inhibited and the ERα is degraded in BC cells.

Therefore, it is critical to understand in what case OU (or possibly other cardiac glycosides) can be administered in women with ERα-positive BC. In turn, present results suggest that the treatment of patients with OU or possibly other CGs could be pursued only after the molecular characterization of breast tumors has evidenced a high expression of ATP1A3 and a low expression of ATP1A1 and ATP1B1. Accordingly, the sensitivity of ERα-positive BC cell lines to OU increases when the cells express high levels of the ATP1B3 mRNA, but not that of the other analyzed Na/K ATPase isoforms.

Present and previous results [4] demonstrate that OU and other CGs have anti-proliferative activities in ERα-positive BC cells at doses compatible with those achieved in the bloodstream of patients treated with these drugs [27]. However, it is well known that OU and other CGs used as medicaments for cardiac arrhythmias work at a concentration, which is close to the lethal one [13]. In turn, it is strongly desirable to reduce their dose of administration by co-treating patients also with the drugs used for BC treatment. We have previously shown that OU displays a synergic anti-proliferative effect with Tam in LumA BC cells and ascribed this phenomenon to the fact that while Tam inhibits the ERα activities, OU induces receptor degradation [4]. Thus, these combined synergic molecular actions allow us to scale down the doses of both drugs to reach a more effective anti-tumor activity. Here, we observed that the sensitivity to OU increases in those ERα-positive BC cell lines also show a high sensitivity to gefitinib.

Although we did not perform a detailed synergistic analysis among ET drugs, HER2 inhibitors, and OU, the data extrapolated by several ERα-positive BC cell lines suggest not only that OU could be co-administered with gefitinib, but also that the inhibition of HER2 activity could synergize with the receptor degradation for the effective reduction in BC cell proliferation, thus implying a molecular circuitry linking HER2 to ERα stability. Notably, an inverse correlation has been observed between fulvestrant and OU as well as letrozole and OU, suggesting these drug combinations could not work for ERα-positive BC treatment. However, these observations further indicate that OU administration alone could be effective when a reduced sensitivity to fulvestrant or letrozole is scored.

In conclusion, the data reported here confirm that OU works as an “anti-estrogen-like” drug in all the ERα-positive BC subtypes (i.e., LumA, LumB, and metastatic BC) and further provide evidence that the use of OU and possibly of other CGs could be indicated for women with ERα-positive tumors expressing high levels of ATP1B3 and low levels of ATP1A1 and ATP1B1. Therefore, we suggest that the ATP1B3 Na/K ATPase isoform could be considered a novel putative biomarker to educate the personalized treatment of ERα-positive BC with OU.

## 4. Materials and Methods

### 4.1. Cell Culture and Reagents

BT-474 and MDAMB231 were purchased by ATCC (Manassas, VA, USA). DMEM (with and without phenol red) and fetal calf serum were purchased from Sigma-Aldrich (St. Louis, MO, USA). Bradford protein assay kit, as well as anti-mouse secondary antibodies, were obtained from Bio-Rad (Hercules, CA, USA). Antibodies against ERα (F-10, mouse) and ubiquitin (P4D1, mouse) were obtained from Santa Cruz Biotechnology (Santa Cruz, CA, USA); anti-vinculin (mouse) antibody was purchased from Sigma-Aldrich (St. Louis, MO, USA). Chemiluminescence reagent for Western blotting was obtained from BioRad Laboratories (Hercules, CA, USA). Fulvestrant (i.e., ICI182,780) was purchased by Tocris (USA). All the other products were from Sigma-Aldrich. Analytical- or reagent-grade products were used without further purification. The identity of the BT-474 and MDAMB231 cell lines was verified by STR analysis (BMR Genomics, Italy).

### 4.2. Cell Manipulation for Western Blotting Analyses

Cells were grown in DMEM with phenol red plus 10% fetal calf serum for 24 h and then treated with the indicated compounds at the indicated doses for the indicated periods. After treatment, cells were lysed in Yoss Yarden (YY) buffer (50 mM Hepes (pH 7.5), 10% glycerol, 150 mM NaCl, 1% Triton X-100, 1 mM EDTA and 1 mM EGTA) plus protease and phosphatase inhibitors. Western blot analysis was performed by loading 20–30 µg of protein on SDS-gels. Gels were run, and the proteins were transferred to nitrocellulose membranes with a Turbo-Blot semidry transfer apparatus from Bio-Rad (Hercules, CA, USA). Immunoblotting was carried out by incubating the membranes with 5% milk or bovine serum albumin (60 min), followed by incubation overnight (o.n.) with the indicated antibodies. Secondary antibody incubation was continued for an additional 60 min. Bands were detected using a Chemidoc apparatus from Bio-Rad (Hercules, CA, USA).

### 4.3. Cell Proliferation and Cell Cycle Assays

Growth curves have been performed as previously reported [4,12,28]. For cell cycle analysis, after each treatment, 1 × 10^6^ cells were washed twice with PBS, fixed dropwise with ice-cold ethanol (70%), and rehydrated with PBS. DNA staining was performed by incubating cells for 30 min at 37 °C in PBS containing 0.18 mg/mL propidium iodide (PI) and 0.4 mg/mL DNase-free RNase (type 1-A). Samples were acquired with a CytoFlex Flow Cytometer (Beckman Coulter) equipped with 488 nm and 635 nm laser sources. Cell cycle analysis was performed using CytExpert v.2.4 software (Beckman Coulter). Doublet discrimination was performed by an electronic gate on FL2-Area vs. FL2-Height parameters.

### 4.4. Measurement of Proteasome Activity

Proteasome activity was measured by employing the Proteasome-Glo™ Assay Kit for chymotrypsin-, trypsin-, and caspase-like activities, purchased from Promega (Madison, MA, USA). Briefly, for measurements of proteasome activity in cells, BT-474 cells were seeded in 96-well plates (10,000 cells/well; each condition in triplicates) in the growth medium, for 24 h. Next, cells were treated with different doses of ouabain or equal quantities of vehicle for an additional 24 h. Notably, the proteasome inhibitor Mg-132 (1 µM) was used as an internal control for all assays. After treatment, the 3 activities of the proteasome were measured according to the manufacturer’s instructions, in a Tecan-Spark Elisa reader every other 30 s, for a total period of 30 min.

### 4.5. Measurement of Na/K ATPase Activity

Cells were seeded in 96-well plates (10,000 cells/well; each condition in a quadruplicate) in the growth medium for 24 h. After 24 h, the Na/K ATPase assay was performed as previously reported [29]. Briefly, cells were washed with NaCl (0.9%) and lysed in ddH_2_O. Then, cells were incubated for 10 min at 37 °C in Assay Buffer 2X (36 mM histidine, 36 mM imidazole, 160 mM NaCl, 30 mM KCl, 6 mM MgCl_2_, 0.2 mM EGTA, pH 7.1) in the presence or absence of different doses of ouabain (10^−9^ to 10^−2^ M) or equal quantities of vehicle for an additional 24 h. Next, 1 mM ATP was added to the reaction mixture, and the plates were incubated for 24 h at 37 °C in 5% CO_2_. The following day, the reaction was blocked using 25 µL/well of SDS 5%, and 125 µL/well of colorimetric solution (ammonium molybdate/H_2_SO_4_ followed by ascorbic acid) was added, to measure the concentration of free P_i_ in each sample. The Na/K ATPase activity was derived by subtracting the activity measured in the presence of ouabain from the total activity. Numeric values of absorbance were obtained with a Tecan-Spark Elisa reader, with the wavelength set at 630 nm. All experiments were performed at least in triplicates.

### 4.6. Colony Assay Formation

Colony assay formation was performed by plating 500 cells per well in 60 mm dishes in triplicate for each condition. Twenty-four hours after plating, cells were washed in PBS, and then the drugs were added to cells in a fresh medium. The medium was changed every 3 days. After 3 weeks, cells were stained with crystal violet. Following extensive washes, stained cells were acquired by computer scanning, and retained crystal violet was quantitated by solubilizing cells with 1% SDS and by reading the corresponding absorbance at Tecan Spark Reader at 575 nm.

### 4.7. Statistical Analysis

Statistical analysis was performed using the InStat version 8 software system (GraphPad Software Inc., San Diego, CA, USA). Densitometric analyses were performed using the freeware software Image J by quantifying the band intensity of the protein of interest with respect to the relative loading control band (i.e., vinculin) intensity. The *p* values are given in figure captions. Student *t*-tests and ANOVA were used when appropriate for evaluating statistical differences among samples.

## Figures and Tables

**Figure 1 ijms-23-11102-f001:**
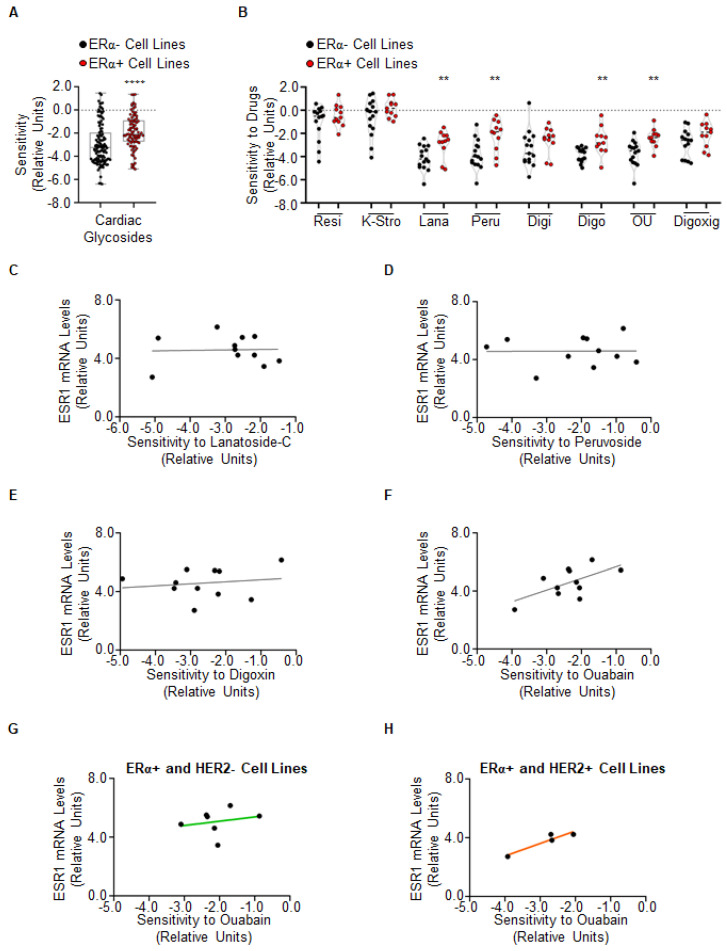
**Analysis of cardiac glycosides in breast cancer cell lines.** Plots in (**A**,**B**) have been generated by using the data in the DepMap portal. Each dot of the plots refers to the value of the specific parameter in a single breast cancer cell line. For details, please see Appendix A. **** (*p* < 0.0001) and ** (*p* < 0.01) indicate significant differences calculated with the Student *t*-test. (**C–F**) Linear regression between sensitivity to the indicated cardiac glycosides and ERα mRNA expression in ERα-positive breast cancer cell lines. (**G**,**H**) Linear regression between sensitivity to ouabain (OU) and ERα mRNA expression in breast cancer cell lines classified as luminal A (LumA-green) or luminal B (LumB-orange). For a detailed explanation of the dataset used, please see the text. For details, please see Appendix A.

**Figure 2 ijms-23-11102-f002:**
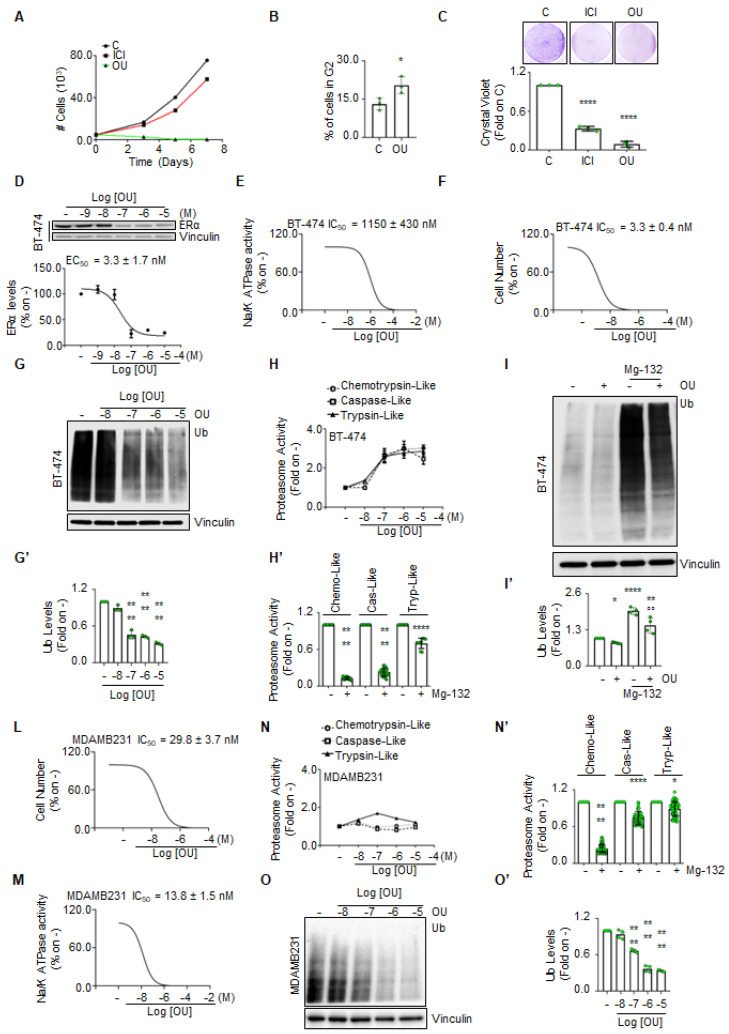
**The impact of ouabain in the regulation of cell proliferation and ERα levels in BT-474 cells.** (**A**) Growth curve analysis in BT-474 cells treated with ouabain (OU-100 nM), and fulvestrant (ICI182,780, ICI-100 nM) for the indicated time points. (**B**) Percentage of the BT-474 cells in the G2 phase of the cell cycle detected after 24 h of ouabain (OU—100 nM) administration. * indicates significant differences, calculated with Student t-test, with respect to C, *p*-value < 0.05. Data are the mean ± the standard deviation of three independent experiments. (**C**) Colony formation assays in BT-474 cells treated for 3 weeks in the presence or the absence of either ouabain (OU-100 nM) or fulvestrant (ICI182,780, ICI-100 nM). **** indicates significant differences, calculated with Anova, with respect to C, *p*-value < 0.0001. Data are the mean ± the standard deviation of three independent experiments. (**D**) Western blot and relative densitometric analyses of ERα levels in BT-474 cells treated for 24 h with the indicated doses of ouabain (OU). (**E**) Na/K ATPase activity and (**F**) number of cells in BT-474 cells treated with ouabain (OU) for 24 h at the indicated doses. (**G**) Western blot and relative densitometric analyses (**G’**) of total ubiquitinated species in BT-474 cells treated for 24 h with the indicated doses of ouabain (OU). ** (*p* < 0.01) indicates significant differences, calculated with Anova, with respect to the-sample. All experiments were performed in triplicate. (**H**) Evaluation of the proteasome activities (i.e., chymotrypsin-like, caspase-like, and trypsin-like) in BT-474 cells, treated for 24 h with the indicated doses of ouabain (OU). (**H’**) The effect of the proteasome inhibitor Mg-132 (1 µM) for 24 h on the three proteasome activities. All experiments were performed twice, in triplicates. (**I**) Western blot analysis and relative densitometric analyses (**I’**) of total ubiquitinated species in BT-474 cells treated for 24 h with ouabain (OU-100 nM) both in the presence or in the absence of the proteasome inhibitor Mg-132 (1 µM). **** (*p* < 0.0001), ** (*p* < 0.01) and * (*p* < 0.05) indicate significant differences, calculated with Anova, with respect to-sample. °° (*p* < 0.01) indicate significant differences, calculated with Anova, with respect to Mg-sample. All experiments were performed in triplicate. (**L**) The number of cells and (**M**) the Na/K ATPase activity in MDAMB231 cells treated with ouabain (OU) for 24 h at the indicated doses. (**N**) Evaluation of the proteasome activities (i.e., chymotrypsin-like, caspase-like, and trypsin-like) in MDAMB231 cells, treated for 24 h with the indicated doses of ouabain (OU). (**N’**) The effect of the proteasome inhibitor Mg-132 (1 µM) for 24 h on the three proteasome activities. All experiments were performed twice, in triplicates. (**O**) Western blot and relative densitometric analyses (**O’**) of total ubiquitinated species in MDAMB231 cells treated for 24 h with the indicated doses of ouabain (OU). ** (*p* < 0.01) indicates significant differences, calculated with Anova, with respect to the-sample. All experiments were performed in triplicate. **** (*p* < 0.0001), and * (*p* < 0.05) indicate significant differences, calculated with Anova, with respect to the-sample. °° All experiments were performed in triplicate. Dots in the bar graph also indicate the number of replicates.

**Figure 3 ijms-23-11102-f003:**
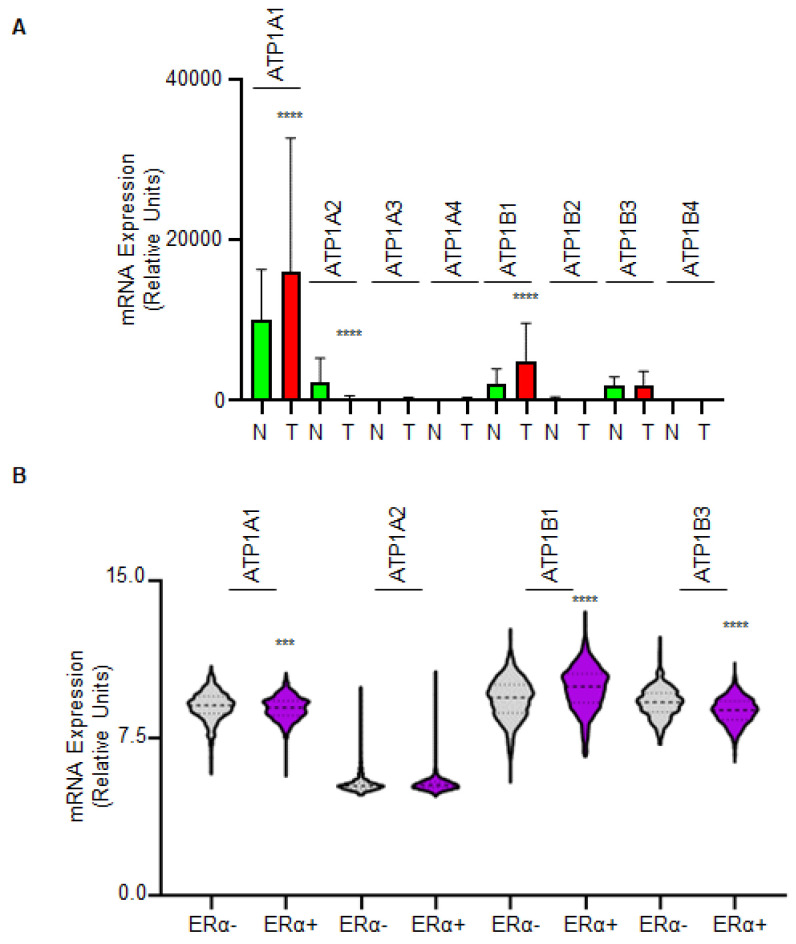
**The expression of the Na/K ATPase isoforms in normal and tumor tissue.** (**A**) RNAseq analyses as retrieved by the https://tnmplot.com/analysis (accessed on 1 July 2022) web portal of the indicated genes in normal (N; the number of samples: 403) and tumor (T; the number of samples: 1097) tissues. Original data are given in Appendix A. (**B**) mRNA expression analyses for the indicated genes as extracted by the https://www.cbioportal.org/ (accessed on 1 July 2022) database and stratified according to estrogen receptor α (ERα) expression. Original data are given in Appendix A. Stratification was done considering the luminal tumors as ERα-positive [1] (ERα+; the number of samples: 1140) and all the others as ERα-negative (ERα-; the number of samples: 758). **** (*p* < 0.0001) and *** (*p* < 0.001) indicate significant differences, calculated with the Student t-test with respect to the N or ERα- sample.

**Figure 4 ijms-23-11102-f004:**
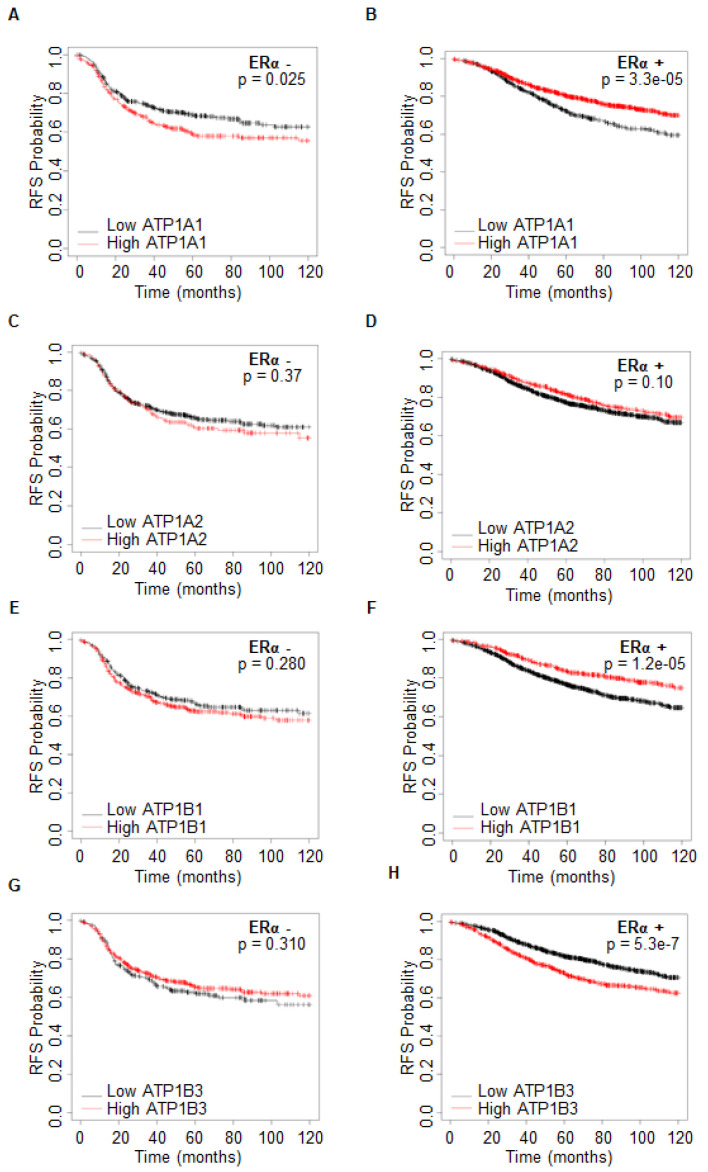
**The expression of Na/K ATPase isoforms in breast cancer progression.** Kaplan–Meier plots show the relapse-free survival (RFS) probability in women carrying ERα-negative (**A**,**C**,**E**,**G**) or ERα-positive (**B**,**D**,**F**,**H**) as a function of the mRNA levels of the indicated genes. All possible cutoff values between the lower and upper quartiles are automatically computed (i.e., auto-select the best cutoff on the website), and the best-performing threshold is used as a cutoff [18]. Details of the parameters of the curves are given in Appendix A. Significant differences between the RFS are given as *p*-value in each panel. All possible cutoff values between the lower and upper quartiles are automatically computed (i.e., auto-select the best cutoff on the website), and the best-performing threshold is used as a cutoff.

**Figure 5 ijms-23-11102-f005:**
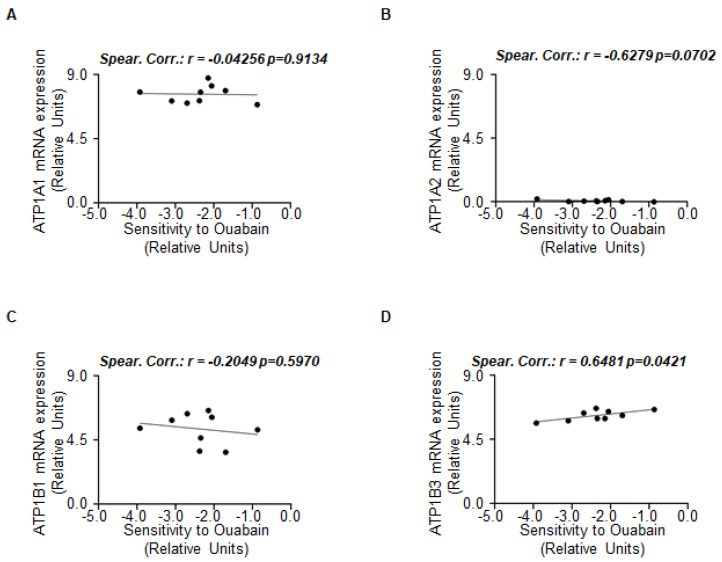
**The correlation between Na/K ATPase isoforms and sensitivity to ouabain in breast cancer cell lines.** Plots have been generated by using the data in the DepMap portal. Each dot of the plots refers to the value of the specific parameter in a single ERα-positive breast cancer cell line. For details, please see Appendix A. (**A**–**D**) Linear regression between sensitivity to ouabain (OU) and the mRNA expression of the indicated genes.

**Figure 6 ijms-23-11102-f006:**
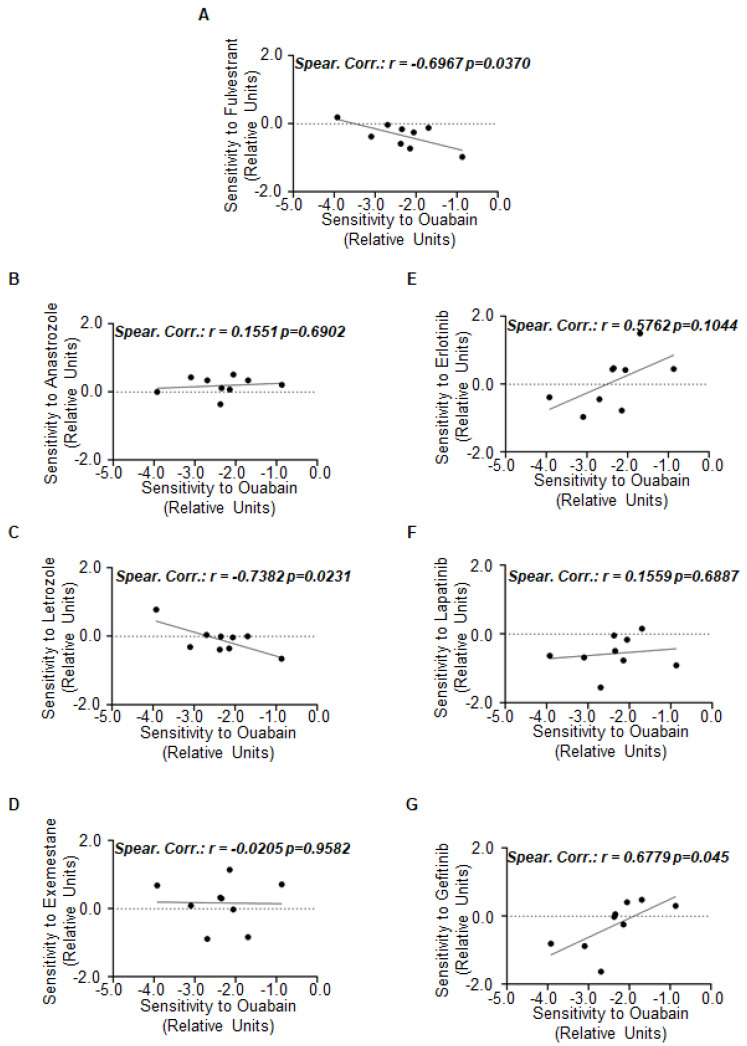
The correlation between the sensitivity to ouabain and the drugs used for breast cancer treatment in breast cancer cell lines. Plots have been generated by using the data in the DepMap portal. Each dot of the plots refers to the value of the specific parameter in a single ERα-positive breast cancer cell line. For details, please see Appendix A. (**A**–**G**) Linear regression between sensitivity to ouabain (OU) and the indicated drugs.

## Data Availability

The data presented in this study are available as Appendix A. Data not included in the Appendix A are available on request to the corresponding author.

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
