# Peer review of "Evaluation of the Sensitivity of Breast Cancer Cell Lines to Cardiac Glycosides Unveils ATP1B3 as a Possible Biomarker for the Personalized Treatment of ERα Expressing Breast Cancers"

_ijms, 2022, doi:10.3390/ijms231911102_

Round 1
Reviewer 1 Report
The present manuscript uses a combination of bioinformatics and experimental data to provide evidence for the use of the cardiac glycoside ouabain (OU, an inhibitor of the Na/K ATPase) as putative drug for treatment of Luminal B (ERα positive and HER2 positive) Breast Cancer (BC). Furthermore, it supports the idea that ATP1B3 could be a biomarker for the personalized treatment of ERα expressing BC. The manuscript presents interesting observations; however, revision is required in order to be considered for publication.
Specific comments:
1. Detailed description in what is presented in Supplementary Tables (1-6) and the corresponding figures is required.
2. Figure 2 is the only one that presents original experimental data obtained from BT-474 cells (model of lumibal B BC, ERα+ and HER2+). In order to strengthen the conclusions, the author should perform the same experiments in a luminal A BC cell line (ERα+ and HER2-). In addition, a representative experiment of those performed for the calculation of IC50s and ED50 values of Figure 3E should be presented.
3. In lines 175-178, the authors’ conclusion ‘Overall, the data reported here indicate that OU prevents cell proliferation and induces ERα degradation in cells modeling the LumB BC phenotype by activating the 26S proteasome, thus demonstrating that OU works in the same manner in both LumA and LumB ERα-positive BC cell lines’ is not supported by the data presented in Figure 2. OU prevents cell proliferation, induces ERα degradation and activates 26S proteasome but it remains to be proven whether the effect on cell proliferation/ERα degradation is due to proteasome activation. The effect of MG-132 on cell proliferation/ERα degradation could be useful.
4. The Kaplan–Meier plots presented in Figure 4 show the relapse-free survival (RFS) probability in women carrying ERα-negative or ERα-positive BC as a function of the mRNA levels of the indicated ATP genes. In order to correlate the RSF with the expression of the indicated genes, all women included in the study should have received the same treatment for their disease. Is this the case?
5. Lines 275-277: ‘Overall, these data suggest that co-administration of OU and gefitinib in ERα positive BC could be a valuable strategy to increase the anti-proliferative activity of both drugs’. The author should test the effect of the combinatorial treatment (OU and gefitinib) in BT-474 cells in order to prove/support the above statement.
Author Response
Reviewer #1
The present manuscript uses a combination of bioinformatics and experimental data to provide evidence for the use of the cardiac glycoside ouabain (OU, an inhibitor of the Na/K ATPase) as putative drug for treatment of Luminal B (ERα positive and HER2 positive) Breast Cancer (BC). Furthermore, it supports the idea that ATP1B3 could be a biomarker for the personalized treatment of ERα expressing BC. The manuscript presents interesting observations; however, revision is required in order to be considered for publication.
Specific comments:
- Detailed description in what is presented in Supplementary Tables (1-6) and the corresponding figures is required.
Author Response: I have now made explicit the content of the supplementary file in the supplementary material section.
- Figure 2 is the only one that presents original experimental data obtained from BT-474 cells (model of lumibal B BC, ERα+ and HER2+). In order to strengthen the conclusions, the author should perform the same experiments in a luminal A BC cell line (ERα+ and HER2-). In addition, a representative experiment of those performed for the calculation of IC50s and ED50 values of Figure 3E should be presented.
Author Response: Regarding the request to show data in luminal A breast cancer cells, these data have been already published in Cancers 2020 Dec 19;12(12):3840. doi: 10.3390/cancers12123840. Please note that in that paper we used two luminal A cell lines (i.e., MCF-7 and ZR-75-1 cells) that model the primary tumor and two luminal A cell lines mimicking the metastatic endocrine therapy resistant tumors (i.e., MCF-7 cells adapted to grow in the presence of 100nM of 4OH-tamoxifen and MCF-7 cells engineered to express the Y537S hyperactive mutant. The representative experiment in figure 2E for ERalpha degradation is already present in figure 2D and D’. I have now shown the IC50 representative experiment for both cell proliferation and Na/K ATPase activity. As per Reviewer #2 request, I also added data in MDAMB231 cells.
- In lines 175-178, the authors’ conclusion ‘Overall, the data reported here indicate that OU prevents cell proliferation and induces ERα degradation in cells modeling the LumB BC phenotype by activating the 26S proteasome, thus demonstrating that OU works in the same manner in both LumA and LumB ERα-positive BC cell lines’ is not supported by the data presented in Figure 2. OU prevents cell proliferation, induces ERα degradation and activates 26S proteasome but it remains to be proven whether the effect on cell proliferation/ERαdegradation is due to proteasome activation. The effect of MG-132 on cell proliferation/ERα degradation could be useful.
Author Response: Please see the response to the comment #2 regarding the effect of OU in luminal A cells. I agree with this Reviewer that the sentence as written is misleading, thus I modified the statement in this way: Overall, the data reported here indicate that OU activates the 26S proteasome, prevents cell proliferation and induces ERα degradation in cells modeling the LumB BC phenotype, thus demonstrating that OU works in LumB ERα-positive BC cell lines as we already reported in two LumA BC cell lines (i.e., MCF-7 and ZR-75-1 cells). Regarding the study of the effect of MG-132 on cell proliferation/ERα degradation we thank this Reviewer but these experiments are out of the scope of the present work.
- The Kaplan–Meier plots presented in Figure 4 show the relapse-free survival (RFS) probability in women carrying ERα-negative or ERα-positive BC as a function of the mRNA levels of the indicated ATP genes. In order to correlate the RSF with the expression of the indicated genes, all women included in the study should have received the same treatment for their disease. Is this the case?
Author Response: The data have been downloaded by the Kaplan-Meier Plotter database. The original analyses have been done for all the women using the same parameters as the tumors included in the database are always the same. Therefore, the treatment status is homogeneous for each gene stratified in the different ways. The original data are as downloaded by the database are available in the supplementary material and the type of analysis has been made clear in the supplementary material section in the text. I have also added explanations of the cut offs in the figure legends.
- Lines 275-277: ‘Overall, these data suggest that co-administration of OU and gefitinib in ERα positive BC could be a valuable strategy to increase the anti-proliferative activity of both drugs’. The author should test the effect of the combinatorial treatment (OU and gefitinib) in BT-474 cells in order to prove/support the above statement.
Author Response: I thank the Reviewer for this comment. The aim of the analyses presented in figure 6 was to understand if OU could synergize with endocrine therapy drugs. Therefore, the experimentally evaluation of this issue is out of the scope of the present work as it would require many different cell lines in which different combinations of drugs is used. The data presented here provide a suggestion for such in silico observation. Nonetheless, I agree with this Reviewer that the sentence is overstated and thus changed it in the following way: ‘Overall, these data suggest that selective co-administration of OU with specific ET drugs in ERα-positive BC could provide possible additional strategies for the management of this kind of disease.’
Reviewer 2 Report
The manuscript of Dr. F. Acconcia evaluates the impact of cardiac glycosides (CGs) on breast cancer cell lines in relation to estrogen receptor-alpha (ERα) status of these cells and then investigates a possible role of ATP1B3, a Na/K ATPase isoform, as a biomarker for luminal breast cancer treatment. This work is the continuation of a previous work showing that two CGs, ouabain and digoxin, activate ERα degradation via the proteosome in luminal A and tamoxifen-resistant breast cancer cell lines. The identification of new molecules capable of countering the mechanisms of resistance to hormone therapy is a major challenge and the CGs properties seem to be interesting and promising. This study brings additional information to the first article but remains limited as to the mechanism of action of these molecules. Several questions remain indeed unanswered: Is the impact of CGs on luminal breast cancer proliferation mediated by ERα degradation alone knowing that proteasome activation by GCs probably affect other proteins? And if the effect is specifically mediated by ERα, by what mechanism, knowing that these molecules do not bind directly the receptor?
Recommendations:
1- The first part of the study is to evaluate the impact that different CGs could have on the proliferation of breast cancer lines by extrapolating from the DepMap portal. The author should explain in a few words the principle of this analysis for readers who are not familiar with this computational approach.
2- The author shows that the sensitivity of ERα-positive breast cancer cell lines to ouabain is linearly correlated with ERα mRNA expression with a more obvious correlation for luminal B tumors. Because luminal B breast cancers generally have less ERα than luminal A breast cancers, the author should discuss this observation.
3- In Figure 2, the author extended previous works performed on luminal A breast cancer cell lines to the luminal B breast cancer cell line, BT-474 and shows similar sensitivity and dynamics to ouabain between both luminal types. As a control, the author should also carry out growth, colony formation and proteasome activity assays on mesenchymal and/or basal-like (ER-negative) breast cancer cell lines in the presence of increased ouabain concentration. This point is important for the understanding of the underlying mechanism of CGs.
4- The author then focuses on the impact of Na/K ATPase isoforms in breast cancer progression, justifying this by the fact that Na/K ATPase is the pharmacological target of ouabain and other CGs. However, the author clearly shows in this and the previous paper that the effect of GCs on ER degradation and on the proliferation of luminal breast cancer cell lines is independent of Na/K ATPase, the EC50 of ouabain on ERα degradation and on proliferation being much lower than its EC50 on Na/K ATPase activity. It is therefore difficult to understand the reason of the study. Although still potentially interesting, the work on Na/K ATPase isoforms should be better argued.
5- In figure 3, statistical analysis must be performed.
Author Response
Reviewer #2
The manuscript of Dr. F. Acconcia evaluates the impact of cardiac glycosides (CGs) on breast cancer cell lines in relation to estrogen receptor-alpha (ERα) status of these cells and then investigates a possible role of ATP1B3, a Na/K ATPase isoform, as a biomarker for luminal breast cancer treatment. This work is the continuation of a previous work showing that two CGs, ouabain and digoxin, activate ERα degradation via the proteosome in luminal A and tamoxifen-resistant breast cancer cell lines. The identification of new molecules capable of countering the mechanisms of resistance to hormone therapy is a major challenge and the CGs properties seem to be interesting and promising. This study brings additional information to the first article but remains limited as to the mechanism of action of these molecules. Several questions remain indeed unanswered: Is the impact of CGs on luminal breast cancer proliferation mediated by ERα degradation alone knowing that proteasome activation by GCs probably affect other proteins?
Author Response: Probably not. What I believe it is happening is that ERα-positive BC cell lines are more sensitive to the anti-proliferative effects of OU and other cardiac glycosides because they also induce receptor degradation. In the discussion section we in fact wrote that ‘we report that ERα-positive BC cell lines are overall more sensitive to the anti-proliferative effects of CGs than the ERα-negative cell lines, most likely because these compounds can activate the 26S proteasome and induce the ERα degradation thus increasing their anti-proliferative activity in ERα-positive BC cell lines’. The addition of triple negative cell lines to the present manuscript further supported this statements.
And if the effect is specifically mediated by ERα, by what mechanism, knowing that these molecules do not bind directly the receptor?
Author Response: The ERα is a transcription factor that in the absence of any ligand regulates the G1 to S phase transition in BC cells. Indeed, it has been known long since that the depletion of ERα induces a blockade in the G1 phase of the cell cycle. Therefore, this nuclear receptor is required for the survival those BC cells that express it. Consequently, the endocrine therapy aims to block either receptor activity or to induce its pharmacological removal from BC cells. In this way, the tumor cells block in G1 phase of the cell cycle and die. Unfortunately, patients treated with the endocrine therapy drugs develop resistance and relapse in a metastatic disease. Often, the mechanisms through which the BC cells become resistant to endocrine therapy drug is the clonal selection of tumor cell expressing a hyperactive receptor variant (e.g., Y537S) that confer estrogen insensitivity and sustain uncontrolled cell proliferation. However, these metastatic cells are STILL addicted to the estrogen receptor survival and proliferative signals. Indeed, if the receptor is removed from such cells, they die.
This is exactly the point of our research activity that we pursued in the last years, as I strongly believe that molecules that induce ERα degradation through various mechanisms can definitively be considered as ‘anti-estrogen-like’ drugs (please see Mol Cell Endocrinol. 2019 Jan 15;480:107-121. doi: 10.1016/j.mce.2018.10.020. Epub 2018 Oct 31). In the case of cardiac glycosides, the additional anti-proliferative effects in ERα-positive BC cells with respect to ERα-negative BC cells is given by the fact that they activate the proteasome and therefore eliminate the receptor-dependent proliferative and survival signals.
Recommendations:
1-The first part of the study is to evaluate the impact that different CGs could have on the proliferation of breast cancer lines by extrapolating from the DepMap portal. The author should explain in a few words the principle of this analysis for readers who are not familiar with this computational approach.
Author Response: I have added the requested sentence at the beginning of the result section 2.1. For this Reviewer’s convenience I am also attaching it here: This free web-based database contains experimental data regarding the profiling of several cancer cell lines of different parameters including drug sensitivity and other several omics data, such as the expression and protein array data, among others.
2-The author shows that the sensitivity of ERα-positive breast cancer cell lines to ouabain is linearly correlated with ERα mRNA expression with a more obvious correlation for luminal B tumors. Because luminal B breast cancers generally have less ERα than luminal A breast cancers, the author should discuss this observation.
Author Response: I have now added the requested discussion in the discussion section.
3-In Figure 2, the author extended previous works performed on luminal A breast cancer cell lines to the luminal B breast cancer cell line, BT-474 and shows similar sensitivity and dynamics to ouabain between both luminal types. As a control, the author should also carry out growth, colony formation and proteasome activity assays on mesenchymal and/or basal-like (ER-negative) breast cancer cell lines in the presence of increased ouabain concentration. This point is important for the understanding of the underlying mechanism of CGs.
Author Response: we agree with this reviewer and thank her/him for this suggestion. We have now added the data regarding the sensitivity to OU of the triple negative cell lines (i.e., basal) MDAMB231. In these cell lines we performed dose response experiments regarding the growth curves, Na/K ATPase activity, proteasome activity and the effect of OU on total ubiquitinated species.
4-The author then focuses on the impact of Na/K ATPase isoforms in breast cancer progression, justifying this by the fact that Na/K ATPase is the pharmacological target of ouabain and other CGs. However, the author clearly shows in this and the previous paper that the effect of GCs on ER degradation and on the proliferation of luminal breast cancer cell lines is independent of Na/K ATPase, the EC50 of ouabain on ERα degradation and on proliferation being much lower than its EC50 on Na/K ATPase activity. It is therefore difficult to understand the reason of the study. Although still potentially interesting, the work on Na/K ATPase isoforms should be better argued.
Author Response: I agree with this comment. However, I have to point out that past and present data indicate that OU, IN ADDITION TO the inhibition of the Na/K ATPase, has the ability to induce the hyperactivation of the proteasome that acts by inducing the ERα degradation. These two effects occur in parallel at high doses of OU in cancer cells while at low doses of the cardiac glycoside the induction of receptor degradation prevails on the inhibition of the Na/K ATPase activity. In turn, the present data do not indicate that the OU pharmacological target is different from the Na/KATPase rather they suggest that this drug has also the novel effect of the hyperactivation of the proteasome at least in the ERα positive BC cells. Therefore, ERα positive BC cells have a higher sensitivity to OU and possibly other cardiac glycosides than ERα negative BC cells. According to the pharmacological indications for human administration of OU, this glycoside is administered by slow intravenous infusion at a loading dose of 15µg/kg to be divided into 3 times at 2-hour intervals (Please see the references included in the revised version of the manuscript) as OU has a very low gastrointestinal absorption rate and rapid excretion rate. Considering the OU molecular weight and a standard human being of 70 Kg and 5 L of blood, these doses correspond to about 0.5 microM OU for each time. At this dose both the Na/K ATPase is inhibited and the ERα is degraded. Therefore, I believe that understanding in what case OU (or other cardiac glycosides) can be administered in women with ERα -positive BC is critical and can be educated by the evaluation of the women survival as a function of the accepted pharmacological target of the drugs being used.
Please note that part of the present response has been introduced in the discussion section.
5-In figure 3, statistical analysis must be performed.
Author Response: I have now added the requested statistical analysis.
Round 2
Reviewer 1 Report
The manuscript can be accepted after minor revision:
1. The content of Supplementary Table 4 should be described in the supplementary material section in the text.
2. MG-132 instead of Mg-132 should be used.
Author Response
Reviewer #1
- The content of Supplementary Table 4 should be described in the supplementary material section in the text.
I have now improved the description of the Supplementary figure 4 in the corresponding section of the manuscript. I also reported it here for the sake of convenience: the detailed information of the settings used by the automatic software, which originates the plots in the main figures are made explicit in the Supplementary Table 4
- MG-132 instead of Mg-132 should be used.
Author Response: I have now modified the name of the compound accordingly.
Please note that, taking advantage of the Grammarly software, I have also corrceted the minor English mistakes evidenced by this reviewer.
Reviewer 2 Report
see file

Author Response
Reviewer #2
In the revised manuscript, the author has addressed most of my previous comments. The only point is the statistical analysis in Figure 3. The author has added in the legend the information about the statistical analysis but Panels A and B show no analysis!
Author Response: I thank this Reviewer for the inconvenience but for some reasons asterisks did not appear in the revised version. I have now double-checked that this problem is not occurring now.